# Treatment of Wounds That Are Difficult to Heal with Photobiomodulation: A Pilot Study

**DOI:** 10.3390/healthcare13141652

**Published:** 2025-07-09

**Authors:** Sara De Angelis, Alessio Conti, Antonella Di Nunzio, Patrizia Stoppa, Fabiano Zanchi, Valerio Dimonte

**Affiliations:** 1North-West District, Azienda Sanitaria Locale Città di Torino, 10146 Torino, Italy; sara.deangelis@aslcittaditorino.it (S.D.A.); antonella.dinunzio@aslcittaditorino.it (A.D.N.); 2University Center of Excellence on Nephrologic, Rheumatologic, and Rare Diseases, Azienda Sanitaria Locale Città di Torino, 10146 Torino, Italy; 3Department of Clinical and Biological Sciences, University of Torino, 10043 Torino, Italy; 4North-East District, Azienda Sanitaria Locale Città di Torino, 10146 Torino, Italy; patrizia.stoppa@aslcittaditorino.it; 5Continuity of Care Department, Azienda Sanitaria Locale Città di Torino, 10146 Torino, Italy; fabiano.zanchi@aslcittaditorino.it; 6Department of Public Health and Pediatrics, University of Torino, 10043 Torino, Italy; valerio.dimonte@unito.it

**Keywords:** hard-to-heal wounds, photobiomodulation, wound care, nurse-led clinics

## Abstract

**Background/Objectives**: Hard-to-heal wounds are resistant to standard treatments and significantly impact patients’ quality of life and healthcare costs. Photobiomodulation with blue light has shown potential in wound healing, but evidence in wounds persisting for extended periods is limited. This pilot study evaluated the effectiveness of an accelerated photobiomodulation protocol in patients with hard-to-heal wounds in a nurse-led outpatient setting. **Methods**: Eleven patients with venous, lymphatic, diabetic, or mixed etiology wounds, unhealed for at least two years, were recruited from two clinics in the North District of the ASL Città di Torino. Participants received twice-weekly sessions of blue light photobiomodulation (EmoLED™, 400–430 nm lasting 60–120 s) for four weeks, in addition to standard care. The wound area was measured at baseline, week 4, and week 12 using the CutiMed Wound Navigator^®^ Version 2.2.8. The secondary endpoints included pain, wound exudate quantity and quality, and the surrounding skin condition. **Results**: All participants (average wound duration 5.9 years; mean area 13.1 cm^2^, SD ± 14.4) completed the treatment; two were lost at follow-up due to unrelated clinical events. No adverse reactions were reported. At week 4, an area reduction was shown in 9 of 11 wounds (mean: 9.5 cm^2^, SD ± 11.4), though not statistically significant (*p* = 0.240). At week 12, a significant reduction was observed (mean: 7.2 cm^2^, SD ± 13; *p* = 0.04), with a mean percentage area decrease of 40.5%. Significant improvements were also noted in pain levels, exudate characteristics, and surrounding skin conditions over time. **Conclusions**: Accelerated blue light photobiomodulation appears to support long-term wound healing and symptom improvement in patients with hard-to-heal wounds. These findings warrant confirmation in larger, controlled studies.

## 1. Introduction

Hard-to-heal wounds significantly impact all aspects of health for both patients and their families, further compromising overall well-being and life satisfaction [1]. They remain a persistent clinical challenge, as they often do not respond to standard treatments and require prolonged, complex care strategies [2]. These wounds, often referred to in the literature as refractory wounds, are defined as those that “fail to proceed through the normal phases of healing in an orderly and timely manner” [3]. Older adults and individuals with comorbidities are most affected, with arterial, venous, diabetic, and lymphatic etiologies being the most prevalent [4]. The increasing prevalence of non-communicable diseases and an aging global population have brought greater attention to wound care, highlighting the substantial societal burden associated with hard-to-heal wounds [5]. As such, they represent a growing global health concern [6]. A systematic review by Martinengo et al. estimated a point prevalence of approximately 0.2%, and a lifetime prevalence of around 1% as of 2019 [7]. Despite their substantial burden, these wounds remain widely underreported [8].

An additional clinical concern is that hard-to-heal wounds are prone to infections, which can lead to tissue necrosis and chronic inflammation, further complicating the healing process [9]. Despite advances in wound care technology, the management of hard-to-heal wounds remains difficult due to the complex interplay of physiological, environmental, and patient-specific factors that influence healing [9]. Current treatment focuses on reducing inflammation, promoting angiogenesis, and supporting tissue regeneration, but wounds often require long-term, intensive care. As a result, patients frequently experience a significantly reduced quality of life, including persistent pain, psychological distress, and social withdrawal [10]. In particular, issues such as discomfort and the financial burden of treatment can further diminish self-esteem and interfere with employment [11]. Beyond the personal toll, hard-to-heal wounds place a substantial financial burden on healthcare systems. In the United Kingdom, the National Health Service (NHS) spends approximately GBP 3.2 billion annually on unhealed wounds, compared to GBP 2.1 billion for healed wounds [12]. Managing an unhealed venous leg wound costs an estimated GBP 13,500, representing over four times the cost of a healed wound of the same etiology [13]. Wound care also demands extensive healthcare resources, including millions of nurse and GP visits (18.6 million practice nurse visits and 10.9 million community nurse visits in 2012/2013) [12]. In Italy, chronic wounds account for an estimated 6.7% of the national Gross Domestic Product, among the highest in the European Union [14]. Direct and indirect costs, such as medical expenses, lost productivity, and reduced quality of life, further amplify the impact. These figures underscore the need for timely and effective system-level interventions to reduce both clinical and economic burden.

When addressing hard-to-heal wounds, both conventional and innovative treatments are employed to enhance healing and manage complications [15]. Conventional approaches include a range of wound dressings, surgical interventions such as autografts, allografts, and cultured epithelial autografts, as well as the application of growth factors to stimulate tissue regeneration [16]. Among the most widely used conventional dressings are hydrogels, hydro-responsive dressings, and those incorporating octenidine and hyaluronan, all of which provide moisture balance, antimicrobial action, and support for cellular activity [17,18]. Alongside these, innovative therapies are gaining ground in the clinical setting. Nanotherapeutics, which utilize nanoparticles for the targeted delivery of drugs and growth factors, show promise in accelerating healing [15]. Stem cell therapy offers regenerative potential by promoting new tissue formation, while 3D bioprinting and bioengineered skin grafts provide customizable and functional skin substitutes [15]. Additionally, cold plasma treatment and photobiomodulation have emerged as a non-invasive option with antimicrobial and tissue-stimulating effects [15,19]. Together, these approaches reflect a growing shift toward personalized and technology-driven wound care.

One of the emerging therapies in the field of wound care is photobiomodulation, a form of phototherapy that utilizes non-ionizing light sources, including lasers, LEDs, and broadband light, in the visible and infrared spectrum [19,20,21]. Among the various wavelengths explored, red (633 nm) and blue (415 nm) light have shown the most promising therapeutic effects [22]. Blue light, in particular, plays a critical role in modulating the inflammatory phase of wound healing by influencing reactive oxygen species. It promotes granulation tissue formation and angiogenesis and enhances re-epithelialization through the activation of fibroblasts and the stimulation of nitric oxide release [22,23]. Through these mechanisms, photobiomodulation offers a non-invasive, supportive strategy to accelerate tissue repair and improve clinical outcomes in patients with hard-to-heal wounds. Despite a growing body of published research, the use of blue light in promoting wound healing is a relatively new and evolving area of research [19,20,21,24,25,26,27]. Although early findings suggest promising outcomes, such as enhanced re-epithelialization, the modulation of reactive oxygen species, and the stimulation of fibroblast activity, further studies are needed to elucidate its biological effects and optimize treatment parameters.

To date, no study on photobiomodulation has specifically targeted hard-to-heal wounds as a distinct clinical category. Previous research has generally included patients with wounds of relatively recent onset (from 90 days to six months) [28,29,30,31], without clearly focusing on the prolonged, treatment-resistant nature that defines hard-to-heal wounds. Additionally, there has been considerable variability in treatment protocols: some studies administered photobiomodulation once per week over a period of 10–12 weeks [29,30,31], while others implemented accelerated regimens with two treatments per week for wounds present for at least 6 weeks [28]. This inconsistency in inclusion criteria and treatment duration limits the generalizability and applicability of findings to chronic wound care. Evaluating the effectiveness of photobiomodulation delivered through a standardized, accelerated protocol may help establish consistent treatment guidelines for hard-to-heal wounds, ultimately contributing to improved clinical outcomes and enhanced quality of life for affected individuals.

Therefore, the aim of this pilot study is to assess the effectiveness of an accelerated photobiomodulation protocol in patients with hard-to-heal wounds.

## 2. Materials and Methods

This pilot intervention study followed Consolidated Standards of Reporting Trials (CONSORT) guidelines and assessed the effects of an accelerated photobiomodulation treatment in a consecutive sample of outpatients with hard-to-heal wounds.

### 2.1. Setting

The ASL Città di Torino is the local health authority responsible for coordinating and delivering healthcare services throughout the urban area of Turin, Italy. Specifically, the research took place in two nurse-led outpatient wound care clinics located in the North District, which provides services to a population of approximately 400,000 residents. Since 2024, ASL Città di Torino has implemented an integrated wound care network across hospital and community settings, emphasizing nurse-led management supported by specialist consultants. These clinics offer individualized follow-up based on patient needs and ensure the continuity of care through shared electronic medical records. In 2024, the North District managed approximately 500 patients with chronic wounds.

### 2.2. Participants

Participants were recruited from the wound care outpatient services of the North-West District of the ASL Città di Torino. A consecutive sample of individuals with hard-to-heal wound injuries who were under the care of the wound care services and had at least one visit in November 2024 was invited to participate. For the purpose of this pilot study, we adopted an intentionally restrictive operational definition of hard-to-heal wounds, identifying wounds that had not healed for at least two years despite appropriate treatment. This choice aimed to focus on a particularly challenging subgroup of patients with long-standing, treatment-resistant wounds. We acknowledge that broader definitions of hard-to-heal wounds typically refer to lack of healing progress within 4–6 weeks [3,32].

The inclusion and exclusion criteria were the following:

Inclusion criteria

-Patients with hard-to-heal wounds (as defined in this study);-Wounds with venous, lymphatic, diabetic, or mixed etiology (intended as a combination of venous, lymphatic, or arterial components documented in the patient’s clinical history);-Age ≥ 18 years;-Written informed consent provided.

Exclusion criteria

-Age < 18 years;-Active oncological disease;-Hard-to-heal wounds with exclusively arterial etiology;-Presence of signs and symptoms of an active infectious process.

No restrictions were applied based on gender, smoking status, comorbidities, or medication use, as the study aimed to reflect real-world outpatient wound care conditions.

### 2.3. Intervention

Individuals who agreed to participate in the study and provided written informed consent were included in this prospective, single-arm treatment cohort and received photobiomodulation using a blue LED device (EmoLED^TM^, Sesto Fiorentino, Italy), a CE-certified (Class IIa) medical device approved for clinical use in the treatment of skin wounds (CE-resolution G1 099,242 0002 Rev.1/2021) in accordance with European Medical Device Regulation. EmoLED^TM^ consists of six LED light sources emitting blue light with a wavelength between 400 and 430 nm. It delivers a power density of 120 mW/cm^2^ and an energy density of 7.2 J/cm^2^. The continuous radiation emitted is evenly distributed across the treatment area by an integrated optical system.

Following adequate wound bed preparation (cleansing and, eventually, debridement), nurses administered the photobiomodulation treatment with Emo LED^TM^ for 60 s at a distance of 4 cm from the wound surface, targeting each 50 mm-diameter circular sub-area (corresponding to an area of 25 cm^2^). For each session, the number of applications was adjusted according to wound size, with a maximum of two applications per session. In the case of diabetic wounds, the treatment time was extended to 120 s, as specified in the device’s technical instructions.

Participants received treatment twice per week, with sessions scheduled every three days. Wound dressings were changed concurrently, following the standard of care. This included cleansing and/or debridement, the application of a hydrofiber dressing, or a hydrofiber dressing with ionic silver in cases of signs of infection. For venous and mixed leg ulcers, compression bandaging was applied when clinically indicated.

### 2.4. Measures

All participants were systematically evaluated at each treatment session and again at twelve weeks from inclusion, corresponding to two months after the conclusion of the photobiomodulation treatment, to assess the long-term effects. The primary endpoint was wound size reduction, assessed using the CutiMed Wound Navigator^®^ application, Version 2.2.8. This software, installed on two mobile devices used in the wound care outpatient services, enabled nurses to measure both the wound area (in square centimeters) and perimeter after each treatment session [33]. The same software and measuring devices were used consistently throughout the entire treatment process and at follow-up, ensuring standardized wound area assessment.

Secondary endpoints included the assessment of pain levels and wound bed/surrounding skin characteristics. Pain level was subjectively reported by participants using a 0–10 mm Numeric Rating Scale (NRS). The presence of wound exudate was rated by nurses using a four-point Likert scale (0 = absent; 1 = scarce; 2 = moderate; and 3 = abundant), while the quality of exudate was evaluated using another four-point Likert scale (0 = clear; 1 = slightly blood; 2 = bloody; and 4 = purulent). Lastly, the condition of the surrounding skin was assessed using a four-point Likert scale (0 = healthy rose; 1 = slightly reddened; 2 = reddened; and 3 = inflamed).

For each participant, sociodemographic data (age, sex, living situation, and level of independence) and clinical data (presence of comorbidities, current therapy, and body mass index) were collected at baseline. Any adverse reactions to the photobiomodulation treatment were monitored and recorded at each treatment session.

### 2.5. Data Analysis

Data analysis was performed using the SPSS statistical package (version 29.2, IBM Corp., Armonk, NY, USA). Descriptive data are presented as mean ± standard deviation (SD). To evaluate the effectiveness of photobiomodulation, paired comparisons were performed between baseline and the end of the treatment (week four), as well as between baseline and at the end of follow-up (week twelve). Given the small sample size and the potential for deviations from normality, non-parametric Wilcoxon signed-rank tests were used.

To assess changes in secondary endpoints over time, the Friedman test was applied to repeated measurements collected at each treatment session and at follow-up. This non-parametric test for repeated measures was chosen due to the ordinal nature of the data and the potential non-normal distribution of differences across time points. Pairwise comparisons between sessions were further explored using the Wilcoxon signed-rank post hoc tests to identify specific time points with significant changes in pain scores.

### 2.6. Ethical Considerations

Participants received both verbal and written information before providing written informed consent to participate in this pilot study. This project was conducted in accordance with the ethical principles outlined in the Declaration of Helsinki for medical research involving human subjects. Formal approval for the study was obtained from the Medical Directorate of the ASL Città di Torino and the Territorial Ethical Committee. This study was registered retrospectively at ClinicalTrials.gov (Registration ID: NCT07042659), as it was originally designed as an exploratory pilot trial.

## 3. Results

A total of 11 patients met the inclusion criteria during November 2024, and all agreed to participate in the study (Table 1). The mean age was 66 years (SD ± 12.4); only two participants were female, and four lived alone. The mean body mass index (BMI) was 33.7 (SD ± 9.6), with only four participants classified as having a normal weight. Nearly all participants presented with comorbidities and risk factors associated with skin wounds. Specifically, four had cardiovascular disease, three had diabetes, and three were diagnosed with hepatitis C.

Among the eleven hard-to-heal wounds treated, seven had a venous etiology, two were of mixed origin, and one each had a diabetic or lymphatic etiology (Table 2). The average wound duration was 5.9 years (SD ± 3.4), with a mean area of 13.1 cm^2^ (SD ± 14.4). Wound size showed considerable variability, ranging from the smallest (0.85 cm^2^) to the largest (41.14 cm^2^).

All 11 participants completed the eight photobiomodulation sessions over a four-week period. However, two were lost to follow-up at twelve weeks: one participant died due to a cardiovascular event, and the other was hospitalized with pneumonia. By the end of the photobiomodulation treatment (week four), nine hard-to-heal wounds had reduced in area, with a mean wound size of 9.5 cm^2^ (SD ± 11.4) and a mean percentage reduction of 24.6% (SD ± 47.6). However, the difference in wound area between baseline and at the end of biomodulation treatment (week four) was not statistically significant (*p* = 0.140). At the end of follow-up (week twelve), seven hard-to-heal wounds had reduced in area compared to baseline, with a mean wound size of 7.2 cm^2^ (SD ± 13) and a mean percentage reduction of 40.5% (SD ± 45). A statistically significant reduction in wound area was observed at week 12 compared to baseline (*p* = 0.04).

No adverse reactions to the photobiomodulation treatment were reported.

Secondary endpoints included perceived pain, wound exudate quantity and quality, and the condition of the surrounding skin. All variables showed significant variation across the treatment and follow-up period, as assessed by the Friedman test (pain: χ^2^ = 77.78; exudate quantity: χ^2^ = 58.73; exudate quality: χ^2^ = 79.53; surrounding skin: χ^2^ = 40.99; all *p* < 0.001). Post hoc Wilcoxon signed-rank tests identified several significant differences between sessions, particularly between early and later time points. While some participants maintained stable values over time, others experienced gradual reductions in pain and exudate quantity and improvements in exudate quality. A statistically significant improvement in the condition of the surrounding skin was also observed between baseline and mid-treatment, with other comparisons showing trends toward significance.

Figure 1 illustrates the evolution of one wound treated with photobiomodulation at baseline, at week two, at week four, and at the end of follow-up.

## 4. Discussion

This pilot study aimed to evaluate the effectiveness of an accelerated photobiomodulation protocol using blue light in patients with hard-to-heal wounds. Conducted in a real-world, nurse-led outpatient setting, our findings provide preliminary evidence supporting the feasibility and potential clinical benefits of this non-invasive treatment approach for stalled wounds that remain unchanged over extended periods, delivered within a nurse-led outpatient setting. While no significant wound area reduction was observed at the end of the four-week treatment period, a statistically significant reduction was achieved at the twelve-week follow-up. These results suggest that photobiomodulation may contribute to long-term wound healing progression, even if short-term effects appear limited.

The findings of this study are consistent with those reported in previous research on photobiomodulation. Notably, a prior study involving 20 patients with chronic wounds resistant to standard treatments found that 16 experienced a reduction in wound size [28]. That study used a similar accelerated protocol, but included wounds of traumatic origin and with a shorter average duration (mean of two years, ranging from six months to five years). Another study, which employed a longer photobiomodulation protocol (10 weeks), demonstrated a significant reduction in the residual area of chronic wounds treated with blue light compared to untreated control wounds [29]. Although this study benefited from a more rigorous design, using control wounds from the same patients to better isolate the effect of the intervention, the median wound duration was 20.5 months. This limits its generalizability to wounds that meet our definition of hard-to-heal, particularly those persisting for longer durations and resistant to conventional treatments. In addition, a study conducted in 12 individuals with systemic sclerosis and associated skin wounds provided retrospective evidence of treatment effectiveness after eight weeks of weekly photobiomodulation sessions [31]. At the same time, the improvement seen in secondary outcomes in the present study aligns with previously reported findings [28,29,31], reinforcing the role of photobiomodulation in promoting tissue repair and modulating inflammatory responses, which contributes to pain relief. Blue light photobiomodulation acts through the absorption of specific wavelengths by endogenous chromophores, such as Protoporphyrin IX, cytochromes, and flavins, leading to increased mitochondrial activity, ATP production, and the modulation of reactive oxygen species [21,34]. These effects promote the resolution of inflammation and support tissue repair by stimulating fibroblast activation, angiogenesis, and macrophage polarization [21,23]. Clinical studies confirm the efficacy and safety of blue light in the treatment of various chronic and inflammatory skin conditions, including diabetic ulcers, venous leg ulcers, burns, and psoriasis [23,34].

Since previous studies did not assess the long-term effectiveness of blue light treatment in initiating and sustaining tissue regeneration, the follow-up data presented in this investigation provide valuable new insights. Our findings highlight the potential delayed effects of photobiomodulation, suggesting that biological processes activated during treatment, such as cellular repair and tissue remodeling, may continue to promote healing well beyond the active intervention phase. However, it should be emphasized that the outcomes of this pilot study were heterogeneous, with some wounds showing limited or no improvement. Given the small sample size, the absence of a control group, and the observational nature of the study, no definitive conclusions regarding the clinical efficacy of photobiomodulation can be drawn. Nevertheless, these preliminary findings should be considered hypothesis-generating and provide a rationale for future larger controlled studies.

Photobiomodulation was initially developed in the 1960s. However, it was not until 2015 that the North American Association for Light Therapy and the World Association for Laser Therapy formally defined and endorsed photobiomodulation as a therapeutic modality. By consensus, they described photobiomodulation as a non-thermal, non-ionizing light treatment that induces photochemical events across various biological levels, ultimately leading to therapeutic outcomes such as pain and inflammation reduction, immunomodulation, and enhanced wound healing and tissue regeneration [35]. The advent of high-efficiency light-emitting diodes in the 1990s marked a turning point, expanding the application of light-based therapies in biomedical contexts. Since then, the use of LEDs in photobiomodulation has become increasingly prevalent, with numerous studies demonstrating its clinical effectiveness [36]. As a result, photobiomodulation is now being explored across a growing range of medical conditions and specialties, reflecting its broad therapeutic potential [37]. Despite promising results, a lack of consensus remains regarding standardized treatment parameters, such as optimal wavelengths, dosages, and targeted therapeutic outcomes, across the existing body of research [19]. This highlights an urgent need within the wound care community for the development of evidence-based clinical protocols and the implementation of well-designed, methodologically rigorous studies to guide the effective use of photobiomodulation.

Hard-to-heal wounds represent a significant challenge for healthcare systems worldwide. These wounds often fail to improve despite appropriate wound management, due to a combination of intrinsic and extrinsic factors [3]. Such factors include venous and arterial insufficiency, advanced age, diabetes, malnutrition, and infection. In addition, the persistence of a pathological, self-perpetuating inflammatory phase plays a critical role in delaying the wound healing process [38]. Blue light is particularly absorbed by enzymes in the electron transport chain and flavoproteins, promoting anti-inflammatory responses, pain reduction, and tissue regeneration [39]. Inflammation is a necessary phase in the wound healing process and, under normal conditions, should resolve within approximately four weeks [40]. However, in the presence of underlying diseases or specific physical conditions, such as advanced age, nutritional deficiencies, or limited mobility, the inflammatory phase may become prolonged or dysregulated, contributing to wounds’ resistance to treatment [41,42].

These factors may help explain why certain hard-to-heal wounds in our pilot study showed minimal or no response to photobiomodulation. Such variability in outcomes is likely influenced by patient-specific factors, including nutritional status, comorbidities, and medication regimens. Notably, one venous wound that had reduced by nearly one-third at the end of the treatment period showed substantial worsening at the 12-week follow-up. This rebound may indicate a need for a longer treatment duration and suggests the potential benefit of integrating lifestyle modifications into the overall therapeutic approach. A similar hypothesis could be proposed for the two wounds in our sample that worsened over time, including one diabetic ulcer that demonstrated limited response from the initial treatment, underscoring the need for a more comprehensive and individualized approach. Vice versa, a venous wound that had previously shown significant worsening (−81% in the surface area) later showed signs of improvement, suggesting that factors other than photobiomodulation may have contributed to the healing process. In this specific case, the patient had a history of intravenous drug use and hepatitis C, which may have contributed to short-term instability. Notably, no local signs of infection were present, and the positive outcome observed at follow-up supports the hypothesis of a behavioral change enhancing the therapeutic effect. These heterogeneous responses underscore the complexity of managing hard-to-heal wounds and highlight the importance of individualized, patient-centered care. In this context, photobiomodulation may be considered a safe, time-efficient, and nurse-manageable adjunct to standard wound care for selected patients, particularly when incorporated into a personalized, multidisciplinary treatment approach.

Nevertheless, several limitations should be acknowledged. First, the small sample size and pilot nature of the study limit the generalizability of the findings. Second, the absence of a control group precludes the ability to draw causal inferences regarding the effectiveness of photobiomodulation. Third, although the wound area was consistently measured using a validated tool (CutiMed Wound Navigator^®^), no standardized wound assessment scales were employed, which limits comparability with other studies. Fourth, although treatment administration was standardized and delivered by trained nurses, we could not fully control or measure patient-related factors such as adherence to care plans, hygiene practices, or substance use, which may have impacted clinical outcomes. Fifth, no pathological or molecular analyses were performed, limiting the understanding of underlying tissue-level responses to photobiomodulation. Sixth, the heterogeneity of wound etiologies and anatomical locations, particularly among diabetic cases, may have introduced variability in treatment response and limited the comparability of outcomes across subgroups. Furthermore, no data were collected on patient-reported outcomes, such as quality of life, which is known to be affected by chronic wounds. In addition, this study adopted a definition of “hard-to-heal wounds” in a context where terminology and classification remain heterogeneous. The lack of consensus in the literature regarding the definition of “chronic” or “hard-to-heal” wounds has been highlighted in recent reviews and may impact both clinical decision-making and research outcomes [32]. Lastly, follow-up was limited to twelve weeks, which may not be sufficient to evaluate wound recurrence or sustained healing over time. Despite these limitations, the study presents several strengths. It was conducted in a real-world, nurse-led outpatient setting, reflecting the practical applicability of photobiomodulation in routine care. Moreover, the use of a standardized and replicable treatment protocol supports the potential scalability of this intervention for future research and clinical implementation.

Future studies should address these limitations by employing randomized controlled designs, including larger and more diverse patient populations. Incorporating both biological markers and patient-reported outcomes would also be valuable to better elucidate the relationship between the physiological effects of blue light and its perceived impact on patients. Additionally, cost-effectiveness analyses would be valuable to support the broader adoption of photobiomodulation in chronic wound care pathways. Given the substantial clinical and economic burden of hard-to-heal wounds, interventions that are safe, well tolerated, and feasible to deliver in community-based, nurse-led settings hold significant promise for improving patient outcomes and optimizing healthcare resources.

## 5. Conclusions

This pilot study suggests that an accelerated blue light photobiomodulation protocol may be a feasible and well-tolerated adjunctive approach in the management of hard-to-heal wounds. Although further controlled studies are required to assess its clinical efficacy, the feasibility, good tolerability, and preliminary signals of potential benefit observed in this study suggest that photobiomodulation warrants further investigation as a possible adjunctive option in the management of hard-to-heal wounds. Integrating such innovations into standardized care pathways may ultimately contribute to improved patient outcomes and reduced healthcare costs.

## Figures and Tables

**Figure 1 healthcare-13-01652-f001:**
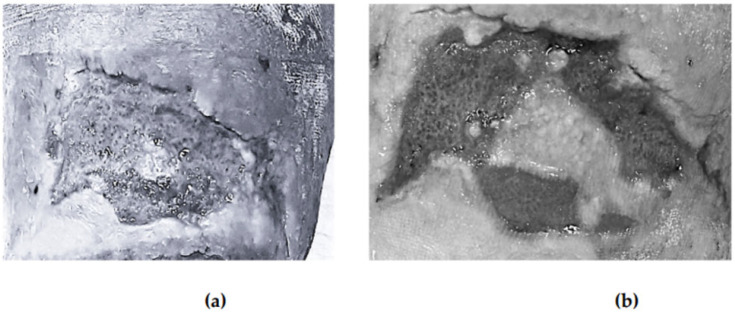
Progression of a venous wound treated with the accelerated photobiomodulation protocol: (**a**) baseline; (**b**) twelve-week follow-up.

**Table 1 healthcare-13-01652-t001:** Characteristics of participants.

#	Age	Sex	BMI	Living with	Comorbidities	Therapies
1	79	Male	23.6	Spouse	Venous insufficiency	Paracetamol
2	57	Male	37.0	Spouse	Hepatitis C	Methadone
3	60	Male	48.5	Alone	Hepilepsy and Obesity	Antiepileptic drugs, diuretics
4	65	Female	39.2	Alone	Hepatitis C	Paracetamol, codeine
5	56	Male	24.4	Father-in-law	Hypertension	Dihydropyridine
6	87	Female	35.1	Alone	Diabetes and Hypertension	Metformin, diuretics, β-blockers
7	82	Male	21.7	Spouse	BPH	α-blockers
8	59	Male	44.0	Partner	Diabetes and Obesity	Metformin
9	47	Male	24.7	Spouse	Hepatitis C	None
10	63	Male	28.1	Partner	Atrial fibrillation	DOACs
11	71	Male	44.0	Alone	Diabetes and Obesity	Metformin, diuretics

BMI: body mass index; BPH: benign prostatic hyperplasia; DOACs: direct oral anticoagulants.

**Table 2 healthcare-13-01652-t002:** Change in areas and delta areas in square centimeters and percentages of hard-to-heal injuries.

#	Etiology (years) Location	Baseline Area cm^2^	Area at W4 cm^2^	Δ Area W4 cm^2^	Δ AREA W4 %	Area at W12 cm^2^	Δ Area W12 cm^2^	Δ Area W12 %
1	Venous (7) Medial malleolus left	32.91	22.54	−10.37	−31.9	42.19	+9.28	+28.19
2	Venous (4) Anterior tibia right	0.85	0.59	−0.26	−30.6	0.24	−0.61	−71.76
3	Venous (4) Anterior tibia left	24.92	5.28	−19.64	−78.8	0	−24.92	−100
4	Mixed (7) Anterior tibia left	41.14	19.97	−21.17	−51.46	/	/	/
5	Venous (9) External malleolus left	3.23	2.35	−0.88	−27.2	0.22	−3.01	−93.19
6	Mixed (5) Posterior tibia left	2.41	1.17	−1.24	−51.45	1.67	−0.74	−30.71
7	Venous (10) Anterior tibia left	3.14	0.94	−2.2	−70.1	1.81	−1.33	−42.36
8	Diabetic (3) Anterior tibia right	3.87	3.66	−0.21	−5.74	4.45	+0.79	+20.41
9	Venous (12) Medial malleolus right	21.89	39.71	+17.82	+81.41	11.47	−10.42	−47.60
10	Venous (2) External tibia right	5.39	2.98	−2.41	−44.71	3.91	−1.48	−27.46
11	Lymphatic (2) Anterior tibia right	4.13	5.77	+1.64	+39.71	/	/	/

cm^2^: square centimeters; **Δ**: delta; W4: week four; W12: week twelve. A negative delta area value indicates a reduction in wound size compared to baseline; positive values indicate an increase.

## Data Availability

The data and materials supporting the findings of this study are available from the corresponding author upon reasonable request due to ethical restrictions related to the protection of participant confidentiality, as approved by the institutional ethics committee.

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
