# Peer review of "Treatment of Wounds That Are Difficult to Heal with Photobiomodulation: A Pilot Study"

_healthcare, 2025, doi:10.3390/healthcare13141652_

Round 1
Reviewer 1 Report
Comments and Suggestions for Authors
This is a manuscript in which I observe several aspects that need to be changed.
Although it is a pilot study, a new treatment is being tested in humans, which is why it is a clinical trial and should be registered with an international clinical trials platform. Another option is to simply describe it as a case series, although in my opinion it is a single-arm clinical trial.
The main negative aspect of the manuscript is that everything is written to support the claim that the therapy works, with limitations, but they mention this. In the results table, I observe that this is not the case. Table 2 shows that only two patients' wounds decreased, while seven patients' wounds increased. Although it may not seem to work, it is difficult to know without a control group. Therefore, the aforementioned conclusion, "This pilot study supports the potential of an accelerated blue light photobiomodulation protocol to improve healing outcomes in patients with hard-to-heal wounds," does not seem adequate or consistent with the data provided.
Author Response
Dear Reviewer,
We sincerely appreciate and have taken all your comments into account; they have enabled us to enhance the overall quality of our manuscript.
A detailed point-by-point response is attached.
We hope you now regard our manuscript as suitable for publication.
Best regards

Reviewer 2 Report
Comments and Suggestions for Authors
Thank you for the opportunity to review your article “Treatment of wounds that are difficult to heal with photobiomodulation: a pilot study ”. Difficult-healing wounds are a nursing and medical problem affecting society worldwide. They primarily affect the elderly, the long-term bedridden and those burdened by multiple diseases. The paper deals with the use of blue light photobiomodulation in the treatment of chronic wounds.
In the reviewer's opinion, it requires several corrections and methodological additions:
- wound etiology - add mixed;
- inclusion and exclusion criterion needs to be clarified,
- whether the EmoLEDä device has quality certification and permission for use in the treatment process,
- by which method the wound areas were measured and whether they were the same measuring tools throughout the treatment process,
- “Our definition of wounds” contradicts the generally accepted definition of a hard-to-heal wound - improve,
Author Response

(The authors gave the same response as above.)

Reviewer 3 Report
Comments and Suggestions for Authors
Please indicate the size of the wounds in the summarry
Please provide data on the location of the non-healing wounds (foot, calf?) surface (anterior, posterior)
As you agree that hard-to-heal wounds are prone to infections, please, provide the microbiological data of the wounds.
Please provide the ankle brachial index in order to know that all patients had compensated blood supply to wounds
Please discuss what happened to patient NO9 as he on the visit W4 had significant increase of the wound size. Do you consider it an adverse reaction to the treatment?
Author Response

(The authors gave the same response as above.)

Reviewer 4 Report
Comments and Suggestions for Authors
Dear authors
I found your article interesting and impressive. Please embedded in the context see my comments.

The language needs edition.
Author Response

(The authors gave the same response as above.)

Round 2
Reviewer 1 Report
Comments and Suggestions for Authors
This is an interesting study, but I see several issues that need to be resolved, aspects that seem to seriously affect the manuscript.
In the affiliations, please include the country where the institutions are located.
The background and methodologies are adequate.
Materials and Methods: This pilot intervention study. Being an intervention study, even if exploratory, it should comply with the guidelines for clinical trials. This appears to be a quasi-experimental, single-arm before-after study: it should be registered in an international clinical trials database and follow CONSORT guidelines. Although they mention that they will submit the registration once obtained, this is a necessary aspect.
Table 2. Change of areas and delta areas in square centimeters and percentages of hard-to-heal injuries
The data do not seem correct. The reduction in area at week 12 does not seem logical. For example, patient 1 had a baseline measurement of 32.91 cm2, and at week 12, it was reported as 42.19 cm2. However, it is stated that there was a change of -9.28 cm2, or -48.58%, or a reduction, when this is not the case. Since this is the primary outcome, and there are few patients, the error of this magnitude makes it difficult to accurately evaluate the results.
Author Response
Dear reviewer, thank you for allowing us to refine our manuscript.
We have carefully considered the matter and provided a point-by-point rebuttal, along with the attached receipt of registration for the study on ClinicalTrials.gov; you can also find the CONSORT checklist at the end of the rebuttal letter.
We hope you are satisfied with the amendments provided and would like to consider the manuscript for acceptance.
Best regards

Reviewer 2 Report
Comments and Suggestions for Authors
Hard-to-heal wounds represent a major and growing challenge for healthcare systems worldwide, particularly aDecting vulnerable populations such as older adults and patients with multiple comorbidities. The additions in the methodological part improve the readability and clarity of the manuscript. The introduction of minimally invasive methods into nursing practice that allow for an improvement and faster wound healing process is a valuable tip. The use of light in wound healing is a beneficial element of therapy. The study should be conducted on a larger number of patients and in other centers.
Author Response
We sincerely thank the reviewer for the encouraging and constructive comments. We are particularly grateful for the recognition of the methodological improvements and for highlighting the potential value of integrating minimally invasive interventions, such as photobiomodulation, into nursing practice to enhance wound care. We fully agree with the reviewer’s observation that hard-to-heal wounds represent a growing challenge, especially among vulnerable populations. As the reviewer noted, the use of light therapy in this context may represent a promising and accessible adjunctive strategy. We also acknowledge and appreciate the suggestion to extend the study to a larger sample and across multiple centers. This is indeed a crucial next step. Our pilot study was designed to assess feasibility and gather preliminary data, and the insights gained will now inform the planning of a larger, multicenter controlled trial aimed at evaluating the clinical effectiveness of this approach more robustly.
Best Regards